**Data Availability Statement:** All relevant data are within the manuscript and its Supporting Information files.

**Funding:** The author(s) received no specific funding for this work.

# Unveiling the urban sports landscape: Profiling participants, motives, and policy implications

Jef Huyghe[1], Nathan D'Hoore[1], Erik Thibaut[1,2], Jeroen Scheerder[1,3]*

1 Departement of Movement Sciences, KU Leuven, Leuven, Belgium, 2 University College Thomas More, Turnhout, Belgium, 3 UNESCO Chair for Sport, Development, Peace (SDP) and Olympic Education, University of the Western Cape, Cape Town, South Africa

* Jeroen.scheerder@kuleuven.be

## Abstract

### Background

The lack of knowledge regarding urban sports poses pressing challenges for governments and sports organisations to deal with in light of its increasing popularity. To develop targeted policy strategies, more insight is needed into the features of urban sports. Therefore, this research aims to establish a profile of urban sports participants in terms of characteristics, behaviours in sports participation, and motives for sports participation.

### Methods

Data was gathered during a large-scale research project on sports participation in the Belgian city of Bruges. A total of 3,951 residents between 6 and 75 years old participated in an online survey. Descriptive statistics and binomial logistic regression analyses were used to examine the differences between urban and traditional sports participants.

### Results

Based on a binomial regression model, the sports level and location significantly predict urban sports participation. The physical motives for sports participation are significantly less important for urban sports participants compared to the group of traditional sports participants. Urban sports participants value the fun and relaxation motives for sports participation significantly more than traditional sports participants.

### Conclusions

This paper aimed to establish a comprehensive profile of urban sports participants and juxtaposed it to that of traditional sports participants. While existing literature often portrays urban sports participants as a relatively homogeneous group, our findings reveal a surprising heterogeneity. Consequently, given its increasing popularity, urban sports present a valuable avenue for governments and sports organisations to engage with a diverse range of sports participants.

**Competing interests:** On behalf of all authors I disclose any competing interests that could be perceived to bias this work.

## Introduction

Since sport is regarded as an important driver of social and economic well-being, it is high on the agenda of local governments and political bodies [1–3]. Specifically, governments–mainly in Western European countries–strive to promote and facilitate sports participation among the non-sporting population and to prevent the interest of sports participants from being lost [4]. By increasing sports participation, governments target societal benefits such as the facilitation of social inclusion and improved physical health and mental health, among other things [5,6]. The Social Return on Investment (SROI) quantifies these aspired societal benefits of sports sector investments. For instance, in England, the SROI ratio was calculated to be 1.91, indicating that every pound invested in sport yielded £1.91 in social benefits [7]. Similarly, in Flanders (Belgium), the SROI ratio in sports is estimated at around 3.5, underscoring the societal value linked to sports participation [8].

However, despite policy efforts to increase sports participation, participation rates have stagnated in recent years [9–11]. At the same time, the existing sports participation is subject to diversification, changing socio-cultural contexts, and social transformation [12]. One of the most notable trends in contemporary sports is the rising popularity of less organised and informal activities. These informal sports have garnered significant interest and now surpass club-organised sports participation rates, which had been the dominant mode for decades [2,13–17]. Moreover, sports behaviour appears to be highly sensitive to societal trends and changes. For example, the recent Covid-19 pandemic had a global impact on sports participation and physical activity. The literature indicates not only a decrease in sports participation but also a significant shift in the organisational setting of sports [3,18,19]. In essence, due to the Covid-19 pandemic, people were obliged to practise sports in an informal and non-organised setting and were therefore forced to use their infrastructure at home or public outdoor urban infrastructure. The Covid-19 pandemic accelerated an existing trend towards, what in the literature has been described as, sport-light, informal, or less-organised sports [17,19].

Alongside and within this trend towards more informal sports settings, since the 1960s and 1970s, there has been the emerging popularity of so-called action sports and lifestyle sports, such as skateboarding, windsurfing, snowboarding, BMX, etc. Action sports and lifestyle sports originated by participants embracing fun, freedom, and fitness and rejecting the traditional and organised aspects of sports [20–24]. In essence, action and lifestyle sports have become more popular and visible over the past five decades as they are experiencing enormous growth in commercialisation with media exposure and sponsorship, and are increasingly part of marketing strategies [23–25]. A prime example of a well-known brand that uses action sports in its promotion is Red Bull. This includes organising major sports events with spectacular tricks in snowboarding, skateboarding, and cliff diving, as well as establishing sponsorship contracts with star athletes in Formula 1, professional cycling, athletics, etc [26–28]. In addition, major events are arising from these action sports blurring the boundaries between entertainment festivals and sporting events [23,29]. This emerging popularity of action and lifestyle sports is also recognised by international sports organisations and institutions. For instance, the International Olympic Committee (IOC) introduced five new action sports, i.e. surfing, skateboarding, 3x3 basketball, sports climbing and BMX freestyle, at the Olympic Games in Tokyo 2020 and has even more action sports scheduled at the Olympics in 2024 and 2028 [30–32].

Consistent with the increasing popularity towards action sports and lifestyle sports is the growing popularity of activities that can be categorised as 'urban sports'. Specifically, urban sports are defined as free and non-organised sports performed in public spaces, often characterised by a less competitive and more social environment compared to traditional sports [33,34]. Typical urban sports are skateboarding, BMX, step, parkour/freerunning, 3x3 basket,

calisthenics, etc. [35]. However, an unequivocal definition of urban sports does not exist and any non-organised sports activity performed in an urban (sport-specific) public space can be regarded as an urban sports activity. For example, football or basketball on a public square or skateboarding in a public park, also referred to as street sports, are also considered urban sports activities [30,33–38]. Additionally, a less demarcated characteristic of urban sports is the so-called 'urban culture' [34,36]. Urban culture is described by Van der Meijde and colleagues (2022) as a term for the close-knit community to which participants in urban sports belong, but also for the 'show' element that is typically an important feature of urban sports [34]. According to Van der Meijde and colleagues, this creative part and the urge to push boundaries are the most important distinguishing characteristics of urban sports compared to more traditional sports activities where competition thrives [30,33,34].

In essence, urban sports fall under the broad scope of non-organised or informal sports and can be considered action or lifestyle sports. However, since general informal sports activities, action sports, or lifestyle sports do not necessarily have to be performed in the public spaces of cities, the category of urban sports activities is mainly relevant for city governments and policymakers. More specifically, despite the absence of participation numbers in urban sports–partly due to the lack of a uniform definition–local governments acknowledge the growing popularity of these activities and increasingly integrate urban sports into their sports policy strategies [30,35,39–44]. For example, in Paris, where parkour originated in the early 1990s, the sport has seen high participation rates, leading the local government to adapt its sport policy strategies by constructing dedicated parkour parks and integrating parkour-friendly features into public spaces [45,46]. Similarly, in Barcelona, the widespread popularity of skateboarding has prompted the city to develop multiple dedicated skateparks and incorporate skateboarding-friendly elements into urban planning, enhancing public spaces for both residents and tourists [47,48].

In sum, policymakers consider urban sports to be highly valuable for cities and their inhabitants. These sports are perceived as catalysts for healthy lifestyles and cooperation due to their recreational, non-competitive, and community-oriented nature [35]. Furthermore, the instrumentalisation of urban sports for societal benefits is deemed particularly effective for several vulnerable groups, such as individuals living in poverty and people with a migration background [22,35,36,49]. Urban sports are especially considered successful in reaching social groups that are generally believed to be difficult to engage, owing to their low cost of participation and equipment, and the accessible locations near cheaper, smaller housing [33,35,50]. Therefore, local governments instrumentalise urban sports to promote integration and well-being, and to increase participation in the welfare state of relatively underrepresented groups [51]. Additionally, urban culture is considered to create new communities of young people, enhancing the city's image and generating economic value by making the city more attractive [22,34–36,49]. This institutionalisation and formalisation of urban sports within the sport landscape is often referred to as 'sportisation' [52,53]. 'Sportisation' specifically refers to the process of incorporating 'play-like' activities into the structure of sports, transforming them into more competitive, regularised, and rationalised endeavours [54,55]. Urban sports are undergoing this organisational development, adopting rules and structures [52,53]. For example, Larsen (2022) delineated the evolution of parkour in Denmark from its origins as a self-organised and play-centric pursuit to a structured discipline marked by weekly training sessions, formal associations, and dedicated sports facilities [52,53].

Yet, despite this 'sportisation' of urban sports, literature is scarce and many questions remain unanswered. The lack of knowledge regarding urban sports poses pressing challenges for local governments to deal with in light of its increasing popularity. More specifically, in order to develop targeted policies to promote and facilitate urban sports, more insight is needed into the profile, preferences and main drivers of those who participate in urban sports.

Therefore, this research aims to investigate (i) the demographic profiles, (ii) the behavioural patterns, and (iii) motivational factors in urban sports, juxtaposed against those prevalent among participants in traditional sports. To examine the difference between the urban sports participant and the traditional sports participant, we draw on data gathered during a larger sports participation study in the city of Bruges (Belgium) [56]. In the following section, we discuss the research context in more depth.

## Material and methods

### Research context

This paper focuses on Belgium, specifically the city of Bruges. Belgium is a rather small country in Western Europe with approximately 11.5 million inhabitants and a strong welfare state [57]. It is a federal state comprising three regions and three communities, divided by language: the Flemish, French, and German-speaking. These communities are each independently responsible for matters such as healthcare, education and culture. Also, sport policy in Belgium falls under the jurisdiction of the three communities separately. Consequently, unlike other federal states, Belgium does not have a national sports policy [57].

Bruges, located in the province of West Flanders, has a population of 118.861 [56]. This study in particular focusses on Bruges, since all data was collected from Bruges citizens. More detailed information on the data collection process is provided in the following section.

### Data collection

The data used in this paper were collected as part of a larger research project on sports participation within the population of the city of Bruges in Flanders. The goal of this study was to acquire insight into the sports and physical activity behaviour of residents in Bruges. For a detailed methodology of the broader research project on sports participation in Bruges, we refer to the work of Scheerder and Huyghe (2023) [56]. The data collection method relevant to this paper is described below.

A total of 25,449 inhabitants of Bruges aged 6 to 75 years were sampled out of an anonymised city register and were contacted by letter to participate in an online survey concerning their participation in sports and physical activity. The inhabitants of Bruges were randomly selected, based on seven demographic characteristics (gender, age, level of education, employment status, nationality, postal code, and family type) to improve representativeness. Qualtrics XM software was used for the survey. The data collection took place from the 6th of June 2022 until the 24th of July 2022. After data collection, weight coefficients were calculated and applied to ensure representativeness with respect to gender, age, and level of education [56]. Ultimately, we obtained a weighted dataset of 3,951 individuals aged 6 to 75 years, representative of the city of Bruges in terms of gender, age and level of education. All data was submitted as part of the paper (S1 File). Before participating in the survey, participants signed a written informed consent, minors were obliged to have consent from their parents, who in that case had to sign the informed consent.

Ethical guidelines were followed and ethical approval for the data collection was given on 25th of April 2022 by the Sociaal-Maatschappelijke Ethische Commissie (SMEC) of the KU Leuven (G-2022-4993-R2(AMD)). An informed consent was signed by all participants.

### Variables

The full survey contained twelve themes of which three were used in this paper: (i) socio-demographic characteristics, (ii) sports participation, and (iii) motives to participate in sports.

**Table 1. Independent variables of sports participation criteria.**

| Variables | Categories | Description |
|---|---|---|
| **Gender** | Men | / |
| | Women | |
| **Age** | 6–12 years old | / |
| | 13–18 years old | |
| | 19–30 years old | |
| | 31–45 years old | |
| | 46–65 years old | |
| | 66–75 years old | |
| **Level of education** | Higher education | Higher professional education or university |
| | Middle education | Secondary education |
| | Lower education | No education or primary education |
| | Still in education | Students |
| **Practised sports** | / | Survey participants were asked to rank their three most practised sports activities. After that, the participants were asked to also list other sports they practised in the past year. |
| **Frequency of sports participation** | Less than once a week (1–27 times per year) | Frequency of sports participation was represented as the number of times per year the participants practised their three main sports. |
| | Once a week (27–52 times per year) | |
| | More than once a week (more than 52 times per year) | |
| **Time per sports session** | Less than one hour per sports session (1–59 min per sports session) | Time per sports session was represented as the average time (in minutes) per sports session. |
| | One hour or more per sports session (60 min or more per sports session) | |
| **Level of sports participation** | In a recreational way only | Level of sports participation was examined by asking the respondents if they practise their sports activities in a 'competitive' way, in a 'recreational' way or in a 'competitive and recreational' way. Resulting in three categories for the level at which the sports are practised. Because only 2.4% of the Bruges sportive population practised sports on a competitive level only, these three categories were recoded into two categories. |
| | In a competitive and recreational way | |
| **Organisational setting of the sports activities** | Organised sports activities only | Organisational setting of the sports activities was queried for each sports activity separately. Participants had thirteen answer options and multiple answer options could be chosen. These were recoded into two variables. |
| | Non- or self-organised sports activities | |
| **Location of sports activities** | In the public space | Location of sports activities was queried for each sports activity separately. Thirteen answer options were possible and multiple answer options could be chosen. These thirteen options were recoded in two new variables. Respondents were allocated as 'in the public space' if they practised their sports in a public park on the street on a public square and/or in the forest/outdoors/nature. |
| | Never in the public space | |
| **Sports companionship** | At least sometimes alone | Sports companionship was measured by asking in which companions the participants usually practise their sports activities. There were seven answer options with different categories of companions, multiple answer options could be chosen. |

Data concerning the latter two themes were only collected among participants who practised sports in the past year.

**Independent variables.** Table 1 shows all the independent variables utilised in this research, except for the motives for sports participation. Socio-demographic characteristics including gender, age, and level of education were considered. In addition, the survey contained detailed questions concerning sports participation, encompassing variables such as (i) practised sports, (ii) frequency of sports participation, (iii) time per sports session, (iv) level of sports participation, (v) organisational setting of sports activities, (vi) location of the sports activities, (vii) sports companionship, and (viii) participants' motives to participate in sports.

**Table 2. Summary of exploratory factor analysis results for the sixteen motives to participate in sports (N = 2,642).**

| Motives to participate in sports | Physical motives | Social motives | Satisfaction and recognition motives | Fun and relaxation motives |
|---|---|---|---|---|
| My health is improving | **0.812** | | | |
| My body becomes more beautiful | **0.755** | | | |
| My physical condition is improving | **0.754** | | | |
| I lose weight | **0.714** | | | |
| I get to know new people | | **0.737** | | |
| I am with my friends | | **0.717** | | |
| It makes me part of society | | **0.707** | | |
| Others stimulated me | | **0.620** | | |
| It is an opportunity to network | | **0.562** | 0.486 | |
| Others look up to me | | | **0.674** | |
| I earn money | | | **0.597** | |
| I feel the kick | | | **0.591** | 0.539 |
| It is a compensation for my hard work-life | | | **0.548** | |
| I can compete | | | **0.540** | 0.452 |
| I have fun | | | | **0.768** |
| I feel less tension, stress, sadness, or aggression | 0.401 | | | **0.509** |
| Eigenvalues | 2.644 | 2.481 | 2.333 | 1.961 |
| % of variance | 16.524 | 15.506 | 14.578 | 12.256 |
| α | 0.782 | 0.760 | 0.678 | 0.471 |

Only factor loadings over 0.40 are shown.

The highest factor loading per motive appears in bold.

Dependent variables: Urban sports participants and non-urban sports participants.

While the operationalisation of the former seven categories is straightforward, we elaborate on the participants' motives. These motives are derived from a scale developed by De Bourdeaudhuij et al. (2005), subsequently updated and validated in Dutch and international literature [58–62]. Based on the above, the current study incorporated sixteen motives for sport participation scored on a Likert scale of 1–7, ranging from 'not applicable to me at all' to 'very applicable to me'.

To streamline variables, a Principal Component Analysis (PCA) with orthogonal rotation (varimax) was conducted on the sixteen items. According to Field's (2009), The Kaiser-Meyer-Olkin (KMO) measure of sampling adequacy (KMO = 0.786) was 'good'. All individual KMO values are higher than 0.690, which is well above the acceptable limit of 0.500 [63]. Given the significant result of Bartlett's test of sphericity (p<0.001), there were sufficient relationships between the variables which makes a factor analysis appropriate. An analysis to obtain eigenvalues for each component showed four components with an eigenvalue higher than one. These four components explained in total 58.86% of the variance. Since the selection of four components was supported by the scree plot, four components were retained for the final analysis.

Table 2 displays the sixteen motives with their rotated factor loadings per cluster, leading to four components: (i) physical motives, (ii) social motives, (iii) satisfaction and recognition motives, and (iv) fun and relaxation motives. Reliability analysis indicated good scores for physical (α = 0.78), social (α = 0.76) and satisfaction and recognition motives (α = 0.68), while fun and relaxation motives showed relatively lower reliability (α = 0.47). Four new variables of clustered motives were created by calculating the average score of the corresponding motives.

Each motive was allocated to the cluster with their highest rotated factor loading which is shown in bold in Table 2.

**Dependent variables: Urban sports participants and non-urban sports participants.** To measure urban sports participation, a new variable was created, as direct inquiry on urban sports participation was absent in the survey. Given the lack of definition for urban sports, as aforementioned, two distinct conceptualisations were employed in this paper.

Firstly, urban sports were identified based on specific sports types. Sixteen sports, drawn from non-exhaustive lists in literature, were classified as urban sports. These sports include 3x3 basket, BMX, breaking, calisthenics, parkour/freerunning, inline/stuntstep, pétanque, skateboard, tricking, bootcamp, boulder, headis, inlineskating, slacklining, tai chi, and urban roundnet (spikeball) [35,64].

Secondly, a broader definition of urban sports was adopted. Individuals engaging in sports activities within non-organised settings and public spaces were categorised as urban sports participants. For example, under this definition, a person playing football with friends in a public park would be considered an urban sports participant.

## Data analysis

SPSS 28.0.1.1 was used for statistical analysis. Initially, descriptive statistics were calculated, and Chi-squares were conducted to test for significant differences between sports participants and non-sports participants, as well as between sports participants in general and urban sports participants.

Subsequently, three binomial regression analyses were conducted. The first regression aimed to get insight into the differences in general population characteristics between the sporting and the non-sporting population in Bruges. The next regressions examined the differences between urban sports participants and (more) traditional (non-urban) sports participants, utilising the two abovementioned definitions of urban sports.

These two binomial regression analyses to examine the differences between urban sports participants and non-urban sports participants contained three blocks. The first block consisted of socio-demographic characteristics (gender, age, educational level), the second block contained sports participation (frequency, time per sports session, level, setting, location, companionship) and the third block included the four motivations to practise sports (physical motives, social motives, satisfaction and recognition motives, and fun and relaxation motives).

In the analysis where urban sports participants were defined by the setting and location of their sports activities, the setting and the location variables were excluded to avoid linearity, as these criteria were integral to the dependent variable's definition.

## Results

### Descriptive statistics

**Total population.** Table 3 presents the descriptive statistics. The weighted dataset contains 3,951 Bruges citizens, with 74.7% reporting sports participation in the past year. Women constitute 50.8% of the total dataset, while men represent 49.2%. Nearly one-third of respondents (31.2%) have a high educational level, and 20.8% are currently in education.

**Sports population.** In the sports-active population, men (51.1%) are slightly more represented than women (48.9%). The highly educated individuals constitute 33.4% of the sports-active population.

The majority of sports participants (69.1%) engage in sports more than once a week, and over half (50.9%) have sports sessions lasting one hour or more. Most participants practise their sports exclusively in a recreational way (70.3%) and in a non-organised setting (73.9%).

**Table 3. Descriptive statistics of the variables.**

| | Total population (N = 3,951) | Sports population (N = 2,950) | Urban sports population (based on specific sports) (N = 171) | Urban sports population (based on location and setting) (N = 755) |
|---|---|---|---|---|
| **Gender (%)** | | | | |
| Men | 49.2 | 51.1 | 60.8* | 53.5 |
| Women | 50.8 | 48.9 | 39.2* | 46.5 |
| **Age (%)** | | | | |
| 6-12y | 8.5 | 10.6 | 12.9 | 9.1 |
| 13-18y | 7.9 | 9.6 | 14.0 | 10.7 |
| 19-30y | 16.8 | 19.9 | 24.6 | 23.9** |
| 31-45y | 21.4 | 23.1 | 23.4 | 22.7 |
| 46-65y | 33.2 | 28.2 | 12.9** | 24.6* |
| 66-75y | 12.1 | 8.6 | 12.3 | 8.9 |
| **Educational level (%)** | | | | |
| High | 31.2 | 33.4 | 28.5 | 31.2 |
| Medium | 28.8 | 26.7 | 22.7 | 25.4 |
| Low | 19.3 | 14.3 | 15.7 | 17.2* |
| Still in education | 20.8 | 25.6 | 33.1* | 26.3 |
| **Sports frequency (%)** | | | | |
| Less than once a week | - | 14.3 | 11.8 | 13.1 |
| Once a week | - | 13.7 | 15.3 | 7.9** |
| More than once a week | - | 69.1 | 72.3 | 78.9** |
| Missing | - | 2.9 | / | / |
| **Time per sports session (%)** | | | | |
| Less than one hour per session | - | 46.0 | 36.7** | 48.2 |
| One hour or more per session | - | 50.9 | 63.3** | 51.8 |
| Missing | - | 3.0 | / | / |
| **Sports level (%)** | | | | |
| Always recreational | - | 70.3 | 58.2** | 69.7 |
| Recreational and/or competitive | - | 26.7 | 41.8** | 30.3 |
| Missing | - | 2.9 | / | / |
| **Sports setting (%)** | | | | |
| Organised only | - | 23.1 | 16.6* | 0.0** |
| Non- /self-organised | - | 73.9 | 83.4* | 100.0** |
| Missing | - | 2.9 | / | / |
| **Sports location (%)** | | | | |
| In public space | - | 25.6 | 52.1** | 100.0** |
| Never in public space | - | 71.4 | 47.9** | 0.0** |
| Missing | - | 2.9 | / | / |
| **Sports companionship (%)** | | | | |
| At least sometimes alone | - | 39.4 | 36.3 | 39.8 |
| Always with companions | - | 57.7 | 63.7 | 60.2 |
| Missing | - | 2.9 | / | / |
| **Motives mean (SD)** | | | | |
| Physical motives | - | 4.74 (1.28) | 4.51 (1.24) | 4.81 (2.22) |
| Social motives | - | 3.26 (1.29) | 3.60 (1.09) | 3.41 (3.31) |
| Satisfaction and recognition motives | - | 2.69 (1.18) | 2.91 (1.18) | 2.85 (1.21) |
| Fun and relaxation motives | - | 5.52 (1.17) | 5.75 (1.03) | 5.61 (1.10) |

*p<0.05

**p<0.01.

A little more than a quarter (25.6%) practice sports in public spaces, and 39.4% of the participants sometimes do so without companionship.

Among the motivational categories, fun and relaxation motives have the highest mean score (5.52), followed by physical motives (4.74), social motives (3.26), and satisfaction and recognition motives (2.69).

**Urban sports population (based on specific sports).** Compared to the overall sports population, urban sports participants–defined by specific sports–are significantly more frequently men (60.8%) and tend to be younger (12.9% aged 6–12 years, 14.0% aged 13–18 years, 24.6% aged 19–30 years). The age group of 46–65 years is significantly underrepresented among urban sports participants.

Urban sports practitioners are significantly more often students (33.1%) and have longer sports sessions (63.3% more than one hour per session). They are also more likely to engage in competitive activities (41.8% competitive and/or recreational) and practise their sports more in public spaces (52.1%) compared to non-urban sports participants. Additionally, urban sports participants significantly more often practise in non-organised settings (83.4%).

**Urban sports population (defined by setting and location of sports participation).** The urban sports participants, defined by the setting and location of their sports activities, show less significant differences in sports participation criteria. However, they practise sports significantly more frequently, with 78.9% participating 'more than once a week' and only 7.9% 'once a week'.

Concerning socio-demographic characteristics, the age group of 19–30 years (23.9%) and individuals with lower educational levels (17.2%) are significantly more represented in the urban sports population, while the age group of 46–65 years (24.6%) is less represented.

## Differences between urban sports participants and (more) traditional sports participants

**Urban sports population (based on specific sports).** Table 4 shows the results of the binomial regression analysis comparing urban sports participants to sports participants who do not engage in urban sports, based on specific sports practices in the past year.

The first model, with general population characteristics, explains a small portion of the variance (Nagelkerke $R^2 = 0.035$). Men ($\exp(\beta) = 1.461$) are significantly more likely to participate in urban sports compared to women. Participants aged 46–65 years are significantly less likely to engage in urban sports compared to the youngest age group (6–12 years). However, the low Nagelkerke $R^2$ indicated limited predictive value from these characteristics alone.

In the second model, shown in Table 4, which includes sports participation criteria, the explanatory power increases (Nagelkerke $R^2 = 0.106$). The age group of 46–65 years remains significantly less likely to participate in urban sports ($\exp(\beta) = 0.316$) than the youngest group, while other general characteristics lose significance.

Sports participation characteristics reveal that urban sports participants are significantly more likely to have sessions lasting one hour or more ($\exp(\beta) = 1.456$) and to engage competitively or recreationally ($\exp(\beta) = 1.638$) rather than solely recreationally. The location of sports activities also shows significant differences: urban sports participants are much more likely to practice in public spaces ($\exp(\beta) = 3.272$) compared to non-urban sports participants. With a Wald statistic of 44.765, the sports location is the most significant predictor of urban sports participation.

In the third model, including motives for sports participation, the Nagelkerke $R^2$ further increases to 0.119. The age group of 46–65 years continues to be significantly less likely to participate in urban sports ($\exp(\beta) = 0.354$) compared to the youngest group. Other general

**Table 4. Binomial logistic regression analysis of urban sports participants and (more traditional) sports participants (N = 2,635).**

| | Model 1: (general population characteristics) | | | Model 2: (sports participation criteria) | | | Model 3: (motives for sports participation | | |
|---|---|---|---|---|---|---|---|---|---|
| | β (S.E.) | Wald | Exp(β) | β (S.E.) | Wald | Exp(β) | β (S.E.) | Wald | Exp(β) |
| **Gender (women = ref)** | | | | | | | | | |
| Men | 0.379* (0.168) | 5.078 | 1.461* | 0.240 (0.177) | 1.845 | 1.272 | 0.225 (0.178) | 1.596 | 1.253 |
| **Age (6-12y = ref)** | | | | | | | | | |
| 13-18y | 0.114 (0.320) | 0.127 | 1.121 | -0.095 (0.335) | 0.080 | 0.909 | 0.028 (0.343) | 0.007 | 1.028 |
| 19-30y | 0.016 (0.378) | 0.002 | 1.016 | -0.182 (0.398) | 0.210 | 0.833 | 0.071 (0.416) | 0.029 | 1.073 |
| 31-45y | -0.345 (0.452) | 0.581 | 0.708 | -0.402 (0.475) | 0.716 | 0.669 | -0.197 (0.490) | 0.162 | 0.821 |
| 46-65y | -1.123*(0.475) | 5.577 | 0.325* | -1.151*(0.498) | 5.344 | 0.316* | -1.039*(0.513) | 4.109 | 0.354* |
| 66-75y | 0.078 (0.490) | 0.025 | 1.081 | 0.035 (0.514) | 0.005 | 1.036 | 0.107 (0.528) | 0.041 | 1.113 |
| **Educational level (high education = ref)** | | | | | | | | | |
| Medium | -0.166 (0.233) | 0.505 | 0.847 | -0.142 (0.237) | 0.358 | 0.868 | -0.157 (0.240) | 0.427 | 0.855 |
| Low | 0.228 (0.279) | 0.664 | 1.256 | 0.058 (0.285) | 0.041 | 1.059 | 0.072 (0.291) | 0.062 | 1.075 |
| Still in education | -0.031 (0.371) | 0.007 | 0.970 | -0.092 (0.381) | 0.059 | 0.912 | -0.149 (0.385) | 0.149 | 0.862 |
| **Frequency (less than once a week = ref)** | - | | | | | | | | |
| Once a week | - | - | - | 0.485 (0.321) | 0.421 | 1.625 | 0.440 (0.324) | 0.423 | 1.553 |
| More than once a week | - | - | - | -0.174 (0.268) | 2.286 | 0.840 | -0.177 (0.272) | 1.851 | 0.838 |
| **Time per sports session (less than one hour per session = ref)** | - | | | | | | | | |
| One hour or more per session | - | - | - | 0.375*(0.182) | 4.232 | 1.456* | 0.343 (0.184) | 3.466 | 1.409 |
| **Level (only in a recreational way = ref)** | - | | | | | | | | |
| In a competitive and a recreational way | - | - | - | 0.493*(0.196) | 6.301 | 1.638* | 0.537*(0.219) | 6.038 | 1.711* |
| **Organisational setting (only organised sports settings = ref)** | - | | | | | | | | |
| At least sometimes non- or self-organised | - | - | - | 0.338 (0.246) | 1.887 | 1.402 | 0.341 (0.246) | 1.922 | 1.406 |
| **Location (never in public space)** | - | | | | | | | | |
| In public space | - | - | - | 1.185**(0.177) | 44.765 | 3.272** | 1.181**(0.178) | 44.190 | 3.256** |
| **Companionship (at least sometimes alone)** | - | | | | | | | | |
| Only in companionship | - | - | - | 0.134 (0.185) | 0.526 | 1.144 | 0.034 (0.189) | 0.033 | 1.035 |
| **Motives for sports participation** | | | | | | | | | |
| Physical motives | - | - | - | - | - | - | -0.170*(0.075) | 5.096 | 0.844* |
| Social motives | - | - | - | - | - | - | 0.146 (0.079) | 3.394 | 1.157 |
| Satisfaction and recognition motives | - | - | - | - | - | - | -0.164 (0.100) | 2.693 | 0.849 |
| Fun and relaxation motives | - | - | - | - | - | - | 0.220*(0.089) | 6.082 | 1.246* |
| **Model summary** | | | | | | | | | |
| Nagelkerke R$^2$ | 0.035 | | | 0.106 | | | 0.119 | | |
| Chi-square | 34.091** | | | 104.268** | | | 118.071** | | |
| -2 Log likelihood | 1171.096 | | | 1100.919 | | | 1087.116 | | |
| Cox & Snell R Square | 0.013 | | | 0.039 | | | 0.044 | | |

β = *correlation coefficient; S.E. = Standard Error; ref = reference category*

Dependent variables: Sports participants practising urban sports (N = 160); sports participants practising other (non-urban) sports (N = 2,475).

*p<0.05

**p<0.01.

population characteristics are insignificant. Differences in session duration no longer significantly contribute to variance. However, sports level and location remain significant predictors, with urban sports participants more likely to engage competitively (exp(β) = 1.711) and practice in public spaces (exp(β) = 3.256).

Motivational factors reveal that physical motives are significantly less important for urban sports participants (exp(β) = 0.844), while fun and relaxation motives are more valued (exp(β)

= 1.246). The sports location remains the highest contributor to the model (Wald = 44.190), followed by fun and relaxation motives (Wald = 6.082), sports level (Wald = 6.038), physical motives (Wald = 5.096), and the age group of 46–65 years (Wald = 4.109).

**Urban sports population (defined by setting and location of sports participation).** The results of the regression analysis using the broader definition of urban sports, based on sports location and the sports setting (N = 755), are not included in this paper. The model showed a Nagelkerke $R^2$ of 0.050 after incorporating general population characteristics, sports participation criteria, and motives, indicating a low predictive value. Additionally, few significant factors for predicting urban sports participation were identified.

## Discussion

The results provide critical insights into the characteristics and motives of urban sports participants in Bruges, Belgium, highlighting significant disparities between urban and traditional sports participants across various demographics (Table 3). Urban sports are predominantly practised by younger individuals and more commonly by men compared to women [34–36,49,65,66] Younger people appear to be drawn to the autonomy of urban sports, where the competition is more about personal progression, and the social aspect of building communities and meeting others [65]

Although the urban sports category generally includes a younger demographic, only the 46–65 age group is significantly underrepresented compared to the traditional sports participants (Table 3). Conversely, men are significantly more represented among urban sports participants (60.8%) compared to the overall sports population (51.1%).

Vossen and Van der Meijde (2024) offer an explanation for these findings, noting that women are less inclined to participate in urban sports due to the hierarchical environment often present, where the most skilled performers dominate [65]. This environment creates a greater barrier for women, who may feel more insecure about their urban sports skills and experience increased pressure from being observed while performing. Older individuals face similar barriers, struggling to achieve the skill level typically required in urban sports. This hierarchical environment also contributes to urban sports being more frequently practised in a competitive manner compared to all sports (Table 3). While not necessarily organised competitions, this hierarchy can be experienced as a form of competition by participants [65]. In sum, the demographic data from this study indicate that urban sports participants are more often men and generally younger compared to more traditional sports participants.

However, the binomial logistic regression model presented in Table 4, shows that factors like gender and educational level have limited predictive value for urban sports participants. Only the age group of 46–65 years contributes to the model's predictive value. This suggests that characteristics such as gender, age, and educational level do not primarily differentiate between urban and non-urban sports participants in our study.

Instead, urban sports participants are distinguished from traditional sports participants by the location and the level of their sports participation, as well as their motives for engaging in sports. This study examines two definitions of urban sports: one based on specific urban sports activities and another broader definition based on the sports' location and setting. Both the descriptive statistics and the binomial regression analysis reveal few differences between traditional sports participants and those defined as urban sports participants under the broader definition. This suggests that the trend towards more informal sports settings has blurred the lines between traditional and urban sports [2,13–17].

More specifically, this trend towards informality aligns with larger shifts in sports participation, where even traditional sports are increasingly practised in non-organised settings.

For instance, our results indicate that recreational cycling, running, and walking as the three most practised sports in the city of Bruges [67]. These activities are frequently conducted in public spaces and self- or non-organised settings, thus qualifying as urban sports under the broader definition. Consequently, defining urban sports solely based on location and setting proves to be insufficient to effectively distinguish between traditional and urban sports.

In contrast, the definition based on specific urban sports seems more suitable for creating a profile of urban sports participants and examining the differences between traditional and urban sports participants. Using this definition, the location of sports emerges as the most distinguishing factor between urban and non-urban sports participants. Our findings substantiate this distinction, with a significantly lower proportion (25.6%) of the total sports population engaging in sports in public locations in comparison to the urban sports population (52.1%). This aligns with the established literature that highlights public spaces as one of the fundamental features of urban sports [33,34]. Surprisingly, the other fundamental feature, a self- or non-organisational setting, does not significantly predict urban sports participation. Moreover, urban sports participants are more likely to engage at competitive or recreational levels compared to solely recreational levels, contradicting prior research that emphasises the social aspect and community in urban sports [33,34].

To reconcile these–at first glance–conflicting findings, we draw on two concepts: *sportisation* and *sportification*. As aforementioned, *sportisation* involves integrating play-like activities into sports organisations, making them more competitive, standardised, and regulated [54,55]. A prime example of *sportisation* is the evolution of parkour from a self-organised and play-like activity to a structured sports discipline with formal associations, and dedicated sports facilities [52,68].

On the other hand, sportification describes the process of non-organised sports evolving into a more institutionalised, formalised, and specialised forms by adding components to increase its appeal [69–74]. The process of *sportification* can be divided into three basic mechanisms: the institutionalisation of practices, the formalisation of standards, and the specialisation of roles [75,76]. The two concepts of *sportisation* and *sportification* are often used interchangeably and ambiguously in the literature, yet they are not the same but often closely intertwined. *Sportisation* involves the transformation of a 'play-like' activity into a sport by integrating it into organised sports structures [52–55]. Conversely, *sportification* refers to the evolution of non-organised sports into more institutionalised, formalised, and specialised forms [69–73,75,76]. These processes are often sequential, with *sportisation* preceding *sportification* as informal activities become organised sports.

In our study, we observed that urban sports in Flanders, including Bruges, undergo *sportisation* and sportification processes. External factors such as media and commercial developments fuel urban sports' *sportisation* [23,26,27,77], with traditional sports organisations and institutions embracing urban sports to enhance their offerings. Traditional sports organisations and institutions use urban sports to enhance the survival, safety, and accessibility of their 'sports branch' [77]. For example, in Flanders, sports federations have begun organising training and competitions for urban sports activities, alongside the development of dedicated urban sports infrastructures [78]. A notable example is the Flemish Gym Federation (GymFed), which incorporated parkour/freerunning into its programme through various competitions, events, and challenges [68]. This integration illustrates the *sportisation* of parkour, as it becomes formally incorporated into the sports landscape of Flanders. Consequently, urban sports are becoming more competitive as they become integrated into traditional sports organisations, reflecting the dynamic nature or urban sports. Furthermore, this integration process implies *sportification*, as organising competitions–a core skill of traditional sport

entities–necessitates the establishment of rules and standardisation. This transition from informal to formalised roles and rules reflects the *sportification* process, which further solidifies the presence or urban sports within the traditional sports landscape.

Put differently, in Flanders, the traditional sports organisations and institutions–such as the Flemish sports administration and sports federations–acknowledge the popularity of urban sports and the shifting sports participation trends, incorporating urban sports to bolster their offerings, a process known as *sportisation*. However, the implementation of these urban sports aligns with the organisational principles of traditional sports organisations [68], mirroring traditional sports in terms of setting and competitiveness, a process referred to as *sportification* [77]. Because of these processes, urban sports are no longer necessarily practised in a self- or non-organised setting and cannot be categorised as less competitive compared to the more traditional sports. Yet, this shift towards competitiveness does not necessarily negate the social aspect and sense of community in urban sports. Nevertheless, further research is necessary to track the evolution of *sportisation* and *sportification* in urban sports.

Lastly, regarding motives for sports participation, two of the four motives significantly contribute to explaining the difference between urban and non-urban sports participants. Urban sports participants prioritise physical motives less and value the fun and relaxation aspects more than non-urban sports participants. This aligns with outlined urban sports' characteristics in literature, emphasising fun, freedom, and physicality while rejecting traditional sports' rigidity [20–24].

It is worth noting that some literature on urban sports also reports the social aspect as a significant motivator for urban sports participants [30,33,34], our study finds social motives to have no significant predictive value in urban sports participation.

In conclusion, our study aimed to construct a comprehensive profile of urban sports participants, yielding findings that diverge from the limited literature available. Specifically, we discovered that traditional sociodemographic characteristics like gender, age, and educational level fail to explain the differences between urban and traditional sports participants. Instead, nuances in sports engagement and underlying motives offer more insight into these differences. Notably, urban sports participants distinguish themselves through their preference for public spaces and non-organised settings.

The evolution towards competitiveness within urban sports hints at a widening appeal, attracting a more heterogeneous group of participants. This evolution stems from the traditionalisation of urban sports, emerging from the *sportisation* and sportification processes discussed earlier. The growing heterogeneous reach serves as a key argument for local governments and sports organisations to invest in urban sports. However, this evolution challenges assumptions of urban sports' inclusivity and attractiveness to young and vulnerable groups [30,33,34,36,37], suggesting a need for further examination and policy consideration.

## Conclusion

The dearth of knowledge concerning urban sports poses pressing challenges for local governments and sports organisations, especially considering its increasing popularity. Developing targeted policies to promote and support urban sports requires a deeper understanding of the characteristics, preferences, and motivations of urban sports participants. Hence, this study sought to compare urban and 'more traditional' sports participants in terms of demographic characteristics, sports participation behaviours, and motivational factors.

Initially rooted in informal, recreational pursuits driven by social and fun motives, urban sports have transitioned into more institutionalised forms characterised by rules and

competitive formats. This evolution reflects the simultaneous influence of two processes: *sportisation* and *sportification*. Rapid integration of urban sports disciplines into the Flemish sports landscape, marked by a surge in organised events and the assimilation of urban sports branches into traditional sports programmes, highlights the transformative nature of these processes.

In essence, these developments are blurring the lines between urban and traditional sports, resulting in limited distinctions between participants of both categories. Our study aimed to delineate a nuanced profile of urban sports participants and contrast it with that of traditional sports participants. While previous literature often depicted urban sports participants as relatively homogeneous, our findings suggest a significant degree of heterogeneity within this demographic. Ultimately, as urban sports continue to gain popularity, they offer local governments and sports organisations an effective avenue to engage with a diverse array of sports participants and potential enthusiasts.

## Practical implications

Firstly, our study challenges the presumption that urban sports are inherently more inclusive. The demographic characteristics of urban sports participants do not significantly differ from those of traditional sports participants. Consequently, integrating urban sports into policy programmes may not address the issue of social inclusion in sports participation, as it appears that the same, yet heterogeneous, groups in society are being reached.

Secondly, the increasing popularity and the heterogeneity of the urban sports participants present an opportunity for (local) governments and sports organisations. Urban sports can be an effective tool to reach a broad spectrum of people, engaging both current and prospective sports participants across various backgrounds and interests.

Thirdly, the rise in popularity of urban sports is closely linked to the trend towards informal sports participation. Governments should consider enhancing and facilitating informal sports settings to encourage those who do not engage with traditional sports organisations or clubs. Fourthly, many urban sports have evolved into more traditional and formalised activities. It is important to recognise that urban sports participants may not necessarily resist traditional sports and their formalised structures. Acknowledging the shift towards organised and competitive formats within urban sports is crucial for accurately understanding their development.

Lastly, the assumption that urban sports environments are less competitive than traditional sports is increasingly inaccurate. Urban sports are frequently practised in competitive contexts, with organised competitions occurring globally.

## Limitations and future research

Firstly, the data of this paper was gathered as part of a larger research project on sports participation in the city of Bruges, which did not include specific questions about urban sports. This limitation restricts the depth and specificity of the information available on urban sports participation. In addition, participants were not able to self-identify as urban sports participants, limiting the accuracy of participant categorisation.

Secondly, the sample had a limited presence of Bruges residents with a migration background and a low socio-economic status. This restriction hindered our ability to thoroughly explore demographic differences between urban and traditional sports participants. Therefore, the findings from this study may not be generalisable to a broader range of cities with more diverse populations. Future research should examine the differences between urban and traditional sports participants, focusing on migration background and socio-economic status.

Understanding these demographic factors can provide a more comprehensive view of urban sports participation.

Thirdly, we utilised two distinct definitions of urban sports based on recent literature to approximate urban sports participation. However, these definitions are neither exhaustive nor conclusive due to the lack of an unequivocal definition in the literature. This could have led to both the misclassification of non-urban sports participants as urban sports participants and the omission of true urban sports participants. Therefore, further research should aim to uncover the traits of various urban sports types rather than generalising the characteristics of urban sports as a whole. This nuanced understanding can help tailor interventions and support for diverse urban sport activities.

Lastly, this study did not gather information on potential barriers to sports participation, which may differ significantly between urban and traditional sports participants. Factors such as violence, weather, and lack of nearby infrastructure were not controlled for, which could influence the findings related to sports participation. Understanding these barriers is crucial to accurately assess and address the needs of both urban and traditional sports participants.

## Supporting information

**S1 File.**
(XLS)

## Acknowledgments

We would like to thank the members of the local sports authority in the city of Bruges for their assistance in data collection, and to the respondents for participating in the online survey.

## Author Contributions

**Conceptualization:** Jef Huyghe, Nathan D'Hoore, Erik Thibaut, Jeroen Scheerder.

**Data curation:** Jef Huyghe.

**Methodology:** Jef Huyghe, Erik Thibaut, Jeroen Scheerder.

**Validation:** Jef Huyghe.

**Visualization:** Jef Huyghe.

**Writing – original draft:** Jef Huyghe.

**Writing – review & editing:** Jef Huyghe, Nathan D'Hoore, Jeroen Scheerder.

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
