## [Decision Letter · Decision Letter 0]

29 Apr 2024

PONE-D-24-02881Unveiling the urban sports landscape: profiling participants, motives, and policy implicationsPLOS ONE

Dear Dr. Huyghe,

Thank you for submitting your manuscript to PLOS ONE. After careful consideration, we feel that it has merit but does not fully meet PLOS ONE’s publication criteria as it currently stands. Therefore, we invite you to submit a revised version of the manuscript that addresses the points raised during the review process. **Please, consider all comments **

We look forward to receiving your revised manuscript.

Kind regards,

Ahmed Mancy Mosa, Ph.D.

Academic Editor

PLOS ONE

Journal Requirements:

3. We are unable to open your Supporting Information file "Urban sport official dataset 8.06.sav". Please kindly revise as necessary and re-upload.

Reviewers' comments:

Reviewer's Responses to Questions

**Comments to the Author**

1. Is the manuscript technically sound, and do the data support the conclusions?

Reviewer #1: Yes

Reviewer #2: Yes

Reviewer #3: Yes

2. Has the statistical analysis been performed appropriately and rigorously? 

Reviewer #1: Yes

Reviewer #2: No

Reviewer #3: Yes

3. Have the authors made all data underlying the findings in their manuscript fully available?

Reviewer #1: Yes

Reviewer #2: Yes

Reviewer #3: Yes

4. Is the manuscript presented in an intelligible fashion and written in standard English?

Reviewer #1: Yes

Reviewer #2: Yes

Reviewer #3: Yes

5. Review Comments to the Author

**Reviewer #1**: Dear authors, first of all, thank you for the submission made to Plos One. The authors seek to establish the profile of urban sports participants regarding their characteristics, behaviors and motivations for practicing sports in a specific region. Overall, the authors did a good job and the conclusions obtained are interesting. Still, there are some points that, if improved, could help the work reach its full potential.

Below you can find my comments in detail.

The introduction is well written and allows a clear understanding of the study problem. The methods are well described and allow replication of the procedures.

Regarding the results, considering and taking into account the typology of the study and the information that is intended to be emphasized, it would make perfect sense in my opinion that there are some graphic elements (figures, diagrams, etc.) that would complement the results.

In addition, taking into account the results, perhaps a point with practical recommendations derived from the findings would make sense. This is because the results can be useful and provide relevant information that can encourage the practice of physical activity.

**Reviewer #2**: The present study aimed to analyse whether sociodemographic characteristics of gender, age, and educational level do explain the differences between urban and traditional sports participants in Brugges, Belgium. Despite being interesting, the paper is very heavy, and becomes exhaustive to read. I consider that the manuscript could be more objective with a reduction of some sections. However, I hope that some of my concerns, that were identified further, after being solved, could contribute to fix it.

Introduction

Despite well designed, this section is longer. Urban sports section should be eliminated and add only a sentence/paragraph about urban sports in this section.

The authors wrote “Although sports participation in Western Europe has increased constantly over the past decades, this trend appears to have stagnated in recent years �7,8,9,10�.” However, bibliographic reference 10 is of 2005.

Methods

Data collection section identified that data were collected in Bruges, however, previous there was indication of the Flanders region. It is important to uniformise all document.

My major concern with this study is related to the methods adopted by the authors to assess the motives. Why didn’t the author use a validated scale? It’s a huge fragility of this work to utilise this methodology. Beyond it, why not provide us with a methodology with more robustness to validate this option?

Results

Table 4 needs to be more descriptive. What means s.e.; ref, per example.

Conclusion

This section has info that should be include in previous.

We need of the practical implications of this study.

**Reviewer #3:** The authors of this article did a great job demonstrating the relevance of their work to society. Furthermore, the article meets the magazine's standards, in line with what the magazine preaches in its values. However, I believe that there are some things that could give more strength and importance to the article and that can be considered or changed by the authors when resubmitting the article for publication. This way, I left my comments in a PDF file of the article, making it easier to specifically locate my considerations in the article.

6. PLOS authors have the option to publish the peer review history of their article (what does this mean?). If published, this will include your full peer review and any attached files.

Reviewer #1: No

Reviewer #2: No

Reviewer #3: **Yes: **Igor Massari Correia

---

## [Author Response · Author response to Decision Letter 0]

11 Jun 2024

We thank the reviewers for their valuable comments on the manuscript and have edited the manuscript to address their concerns and suggestions. Based on their suggestions, we believe the modifications we made enhanced the clarity and depth of the manuscript. All specific adjustments are explained in the rebutal letter we uploaded with the manuscript.

---

## [Decision Letter · Decision Letter 1]

17 Jun 2024

Unveiling the urban sports landscape: profiling participants, motives, and policy implications

PONE-D-24-02881R1

Dear Dr. Huyghe,

We’re pleased to inform you that your manuscript has been judged scientifically suitable for publication and will be formally accepted for publication once it meets all outstanding technical requirements.

Kind regards,

Ahmed Mancy Mosa, Ph.D.

Academic Editor

PLOS ONE

Additional Editor Comments (optional):

Reviewers' comments:

Reviewer's Responses to Questions

**Comments to the Author**

1. If the authors have adequately addressed your comments raised in a previous round of review and you feel that this manuscript is now acceptable for publication, you may indicate that here to bypass the “Comments to the Author” section, enter your conflict of interest statement in the “Confidential to Editor” section, and submit your "Accept" recommendation.

Reviewer #1: All comments have been addressed

Reviewer #2: All comments have been addressed

2. Is the manuscript technically sound, and do the data support the conclusions?

Reviewer #1: Yes

Reviewer #2: Yes

3. Has the statistical analysis been performed appropriately and rigorously? 

Reviewer #1: Yes

Reviewer #2: Yes

4. Have the authors made all data underlying the findings in their manuscript fully available?

Reviewer #1: Yes

Reviewer #2: Yes

5. Is the manuscript presented in an intelligible fashion and written in standard English?

Reviewer #1: Yes

Reviewer #2: Yes

6. Review Comments to the Author

Reviewer #1: Dear authors, thank you very much for your quick and prompt response to the requested revisions. After the changes made, I have no reservations regarding the manuscript and therefore recommend its acceptance.

Reviewer #2: (No Response)

7. PLOS authors have the option to publish the peer review history of their article (what does this mean?). If published, this will include your full peer review and any attached files.

Reviewer #1: No

Reviewer #2: No

---

## [Editor Report · Acceptance letter]

25 Jun 2024

PONE-D-24-02881R1 

PLOS ONE

Dear Dr. Huyghe, 

I'm pleased to inform you that your manuscript has been deemed suitable for publication in PLOS ONE. Congratulations! Your manuscript is now being handed over to our production team.

Kind regards, 

on behalf of

Dr. Ahmed Mancy Mosa 

Academic Editor

PLOS ONE